# CROSS-LINKED VARIATIONAL AUTOENCODERS FOR GENERALIZED ZERO-SHOT LEARNING

**Edgar Schönfeld**
Bosch Center for AI
edgar.schoenfeld@bosch.com

**Sayna Ebrahimi & Trevor Darrell**
Berkeley AI Research
{sayna,trevor}@eecs.berkeley.edu

**Samarth Sinha**
University of Toronto
samarth.sinha@mail.utoronto.ca

**Zeynep Akata**
University of Amsterdam
z.akata@uva.nl

## ABSTRACT

Most approaches in generalized zero-shot learning rely on cross-modal mapping between an image feature space and a class embedding space or on generating artificial image features. However, learning a shared cross-modal embedding by aligning the latent spaces of modality-specific autoencoders is shown to be promising in (generalized) zero-shot learning. While following the same direction, we also take artificial feature generation one step further and propose a model where a shared latent space of image features and class embeddings is learned by aligned variational autoencoders, for the purpose of generating latent features to train a softmax classifier. We evaluate our learned latent features on conventional benchmark datasets and establish a new state of the art on generalized zero-shot learning. Moreover, our results on ImageNet with various zero-shot splits show that our latent features generalize well in large-scale settings

## 1 INTRODUCTION

Generalized zero-shot learning (GZSL) is a classification task where no labeled training examples are available from some of the classes. Many approaches learn a mapping between images and their class embeddings (Frome et al., 2013; Akata et al., 2016; Norouzi et al., 2013; Xian et al., 2016; Akata et al., 2015). For instance, ALE (Akata et al., 2016) maps CNN features of images to a per-class attribute space. An orthogonal approach to GZSL is to augment data by generating artificial image features, such as Xian et al. (2018b) who proposed to generate image features via a conditional WGAN. As a third approach, Tsai et al. (2017) proposed to learn a latent space embedding by transforming both modalities to the latent spaces of autoencoders and match the corresponding distributions by minimizing the Maximum Mean Discrepancy (MMD). Learning such cross-modal embeddings can be beneficial for potential downstream tasks that require multimodal fusion. In this regard, Ramakrishnan et al. (2017) recently used a cross-modal autoencoder to extend visual question answering to previously unseen objects.

Although recent cross-modal autoencoder architectures represent class prototypes in a latent space (Mukherjee et al., 2017; Tsai et al., 2017), better generalization can be achieved if the shared representation space is more amenable to interpolation between different classes. Variational Autoencoders (VAEs) are known for their capability in accurate interpolation between representations in their latent space, i.e. as demonstrated for sentence interpolation (Bowman et al., 2015) and image interpolation (Higgins et al., 2016). Hence, in this work, we train VAEs to encode and decode features from different modalities, and align their latent spaces by matching the parametrized latent distributions and by enforcing a cross-modal reconstruction criterion. Since we learn representations that are oblivious to their origin, a zero-shot visual classifier can be trained using latent space features from semantic data.

Our contributions in this work are as follows. (1) We propose a model that learns shared cross-modal latent representations of multiple data modalities using simple VAEs via distribution alignment and

cross alignment objectives. (2) Our model establishes the new state-of-the-art performance on generalized zero-shot settings on conventional benchmark datasets. (3) Finally, we show that the latent features learned by our model improve the state of the art in the truly large-scale ImageNet dataset in all splits for the generalized zero-shot learning task.

## 2 METHOD

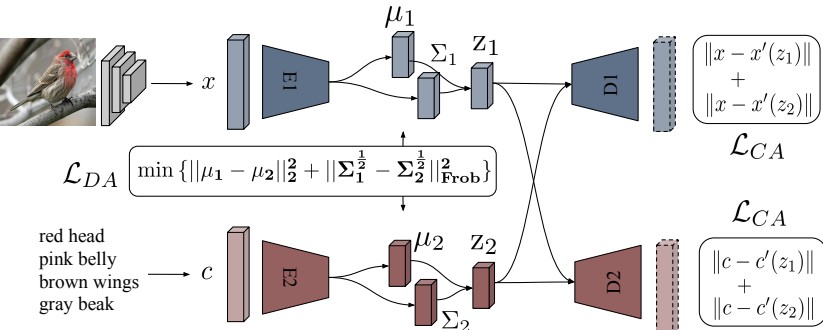

Figure 1: CADA-VAE learns a cross-modal embedding for (generalized) zero-shot classification in the latent space of cross-aligned VAEs, by combining cross-reconstruction and distribution alignment. The upper VAE embeds image features, the lower attributes. The embedding is density-based and allows sampling training examples.

**Generalized Zero-shot Learning** Let $S = \{(x, y, c(y)) | x \in X, y \in Y^S, c(y) \in C\}$ be a set of training examples, consisting of image-features $x$, e.g. extracted by a CNN, class labels $y$ available during training and class-embeddings $c(y)$. Typical class-embeddings are vectors of continuous attributes or Word2Vec features (Mikolov et al., 2013). In addition, an set $U = \{(u, c(u)) | u \in Y^u, c(u) \in C\}$ is used, where $u$ denote unseen classes from a set $Y^u$, which is disjoint from $Y^S$. Here, $C(U) = \{c(u_1), ..., c(u_L)\}$ is the set of class-embeddings of unseen classes. In the legacy challenge of ZSL, the task is to learn a classifier $f_{ZSL} : X \to Y^U$. However, in this work, we focus on the more realistic and challenging setup of generalized zero-shot learning (GZSL) where the aim is to learn a classifier $f_{GZSL} : X \to Y^U \cup Y^S$.

**The Objective Function** `CADA-VAE` is trained with pairs of image features and attribute vectors of seen classes. The data of each pair has to belong to the same class. In this process, an image feature encoder and an attribute encoder learn to transform the training data into the shared latent space. The encoders belong to two VAEs with a common latent space. Once the VAEs are trained, a softmax classifier is trained on both *seen* image data and *unseen* attributes, after they are transformed into the latent representation. As the VAE encoding is non-deterministic, many latent features are sampled for each datapoint. Since we only have one attribute vector per class, we oversample latent-space encoded features of *unseen* classes. To test the classifier, the visual test data is first transformed into the latent space, using only the predicted means $\mu$ of the latent representation. The Objective function for training the VAEs is derived as follows. For every modality $i$ (image features, attributes), a VAE is trained. The basic VAE loss for a feature $x$ of modality $i \in 1, 2, ..M$ is:

$$\mathcal{L}_{\text{basic}} = \sum_i^M \mathbb{E}[\log p(x^{(i)}|z)] - \beta D_{KL}[q(z|x^{(i)})|p(z)] \qquad (1)$$

where $D_{KL}$ represents the Kullback-Leibler Divergence, $\beta$ is a weight, $q(z|x^{(i)}) = \mathcal{N}(\mu, \Sigma)$ is the VAE encoder consisting of a multilayer perceptron, and $p(z)$ is a Gaussian prior. Additionally, each encoded datapoint is decoded into every available modality, e.g. encoded image features are decoded into attributes and vice versa. Consequently, we minimize the L1 cross-reconstruction loss:

$$\mathcal{L}_{CA} = \gamma \sum_i^M \sum_{j \neq i}^M |x^{(j)} - D_j(E_i(x^{(i)}))|. \qquad (2)$$

where $\gamma$ is a weight. The L1 loss empirically proved to provide slighthly better results than L2. Furthermore, the 2-Wasserstein $W$ distance between the multivariate Gaussian latent distribution of image features and attributes is minimized:

$$\mathcal{L}_{DA} = \sum_i^M \sum_{j \neq i}^M W_{ij} \; ; \text{with } W_{ij}^2 = ||\mu_i - \mu_j||_2^2 + ||\Sigma_i^{\frac{1}{2}} - \Sigma_j^{\frac{1}{2}}||_{Frobenius}^2. \tag{3}$$

The VAE is trained using the final objective $\mathcal{L} = \mathcal{L}_{basic} + \mathcal{L}_{CA} + \mathcal{L}_{DA}$. We refer to the Cross-Aligned and Distribution-Aligned VAE as CADA-VAE. In addition, we test the variant $\mathcal{L} = \mathcal{L}_{basic} + \mathcal{L}_{CA}$, termed CA-VAE, and the variant $\mathcal{L} = \mathcal{L}_{basic} + \mathcal{L}_{DA}$, referred to as DA-VAE. A latent size of 64 is used for all experiments, except 128 for ImageNet.

## 3 EXPERIMENTS

We evaluate our framework on zero-shot learning benchmark datasets CUB-200-2011 (Welinder et al., 2010), SUN attribute (Patterson & Hays, 2012), AwA1 and 2 (Lampert et al., 2009; Xian et al., 2018a) for the GZSL setting. All image features used for training the VAEs are extracted from the 2048-dimensional final pooling layer of a ResNet-101. To avoid violating the zero-shot assumption, i.e. test classes need to be disjoint from the classes that ResNet-101 was trained with, we use the proposed training splits in Xian et al. (2018a). As class embeddings, attribute vectors were utilized if available. For ImageNet we used Word2Vec (Mikolov et al., 2013) embeddings provided by Changpinyo et al. (2016). All hyperparameters were chosen on a validation set provided by Xian et al. (2018a). We report the harmonic mean (H) between seen (S) and unseen (U) average per-class accuracy, i.e. the Top-1 accuracy is averaged on a per-class basis.

**Generalized Zero-Shot Learning** We compare our model with 11 state-of-the-art models. Among those, CVAE (Mishra et al., 2017), SE (Verma et al., 2017), and f-CLSWGAN (Xian et al., 2018b) learn to generate artificial visual data and thereby treat the zero-shot problem as a data-augmentation problem. On the other hand, the classic ZSL methods DeViSE (Frome et al., 2013), SJE (Akata et al., 2015), ALE (Akata et al., 2016), EZSL (Romera-Paredes & Torr, 2015) and LATEM (Xian et al., 2016) use a linear compatibility function or other similarity metrics to compare embedded visual and semantic features; CMT (Socher et al., 2013) and LATEM (Xian et al., 2016) utilize multiple neural networks to learn a non-linear embedding; and SYNC (Changpinyo et al., 2016) learns by aligning a class embedding space and a weighted bipartite graph. ReViSE (Tsai et al., 2017) proposes a shared latent manifold learning using an autoencoder between the image features and class attributes.

The results in Table 1 show that our CADA-VAE outperforms all other methods on all datasets. Moreover, our model achieves significant improvements over feature generating models most notably on CUB. Compared to the classic ZSL methods, our method leads to at least $100\%$ improvement in harmonic mean accuracies. In the legacy challenge of ZSL setting, which is hardly realistic, our CADA-VAE provides competitive performance, i.e. 60.4 on CUB, 61.8 on SUN, 62.3 on AWA1, 64.0 on AWA2. However, in this work, we focus on the more practical and challenging GZSL setting. We believe the obtained increase in performance by our model can be explained as follows. CADA-VAE learns a shared representation in a weakly supervised fashion, through a cross-reconstruction objective. Since the latent features have to be decoded into every involved modality, and since every modality encodes complementary information, the model is encouraged to learn an encoding that retains the information contained in all used modalities. In doing so, our method is less biased towards learning the distribution of the seen class image features, which is known as the projection domain shift problem (Fu et al., 2014). As we generate a certain number of latent features per class using non-deterministic encoders, our method is also akin to data-generating approaches. However, the learned representations lie in a lower dimensional space, i.e. only 64, and therefore, are less prone to bias towards the training set of image features. In effect, our training is more stable than the adversarial training schemes used for data generation (Xian et al., 2018b).

**ImageNet Experiments** In Xian et al. (2018a) several evaluation splits were proposed with increasing granularity and size both in terms of the number of classes and the number of images. Note that since all the images of 1K classes are used to train ResNet-101, measuring seen class accuracies would be biased. However, we can still evaluate the accuracy of unseen class images in the GZSL search space that contains both seen and unseen classes. Hence, at test time the $1K$ seen classes

| | CUB | | | SUN | | | AWA1 | | | AWA2 | | |
| Model | **S** | **U** | **H** | **S** | **U** | **H** | **S** | **U** | **H** | **S** | **U** | **H** |
|---|---|---|---|---|---|---|---|---|---|---|---|---|
| CMT Socher et al. (2013) | 49.8 | 7.2 | 12.6 | 21.8 | 8.1 | 11.8 | 87.6 | 0.9 | 1.8 | 90.0 | 0.5 | 1.0 |
| SJE Akata et al. (2015) | 59.2 | 23.5 | 33.6 | 30.5 | 14.7 | 19.8 | 74.6 | 11.3 | 19.6 | 73.9 | 8.0 | 14.4 |
| ALE Akata et al. (2016) | 62.8 | 23.7 | 34.4 | 33.1 | 21.8 | 26.3 | 76.1 | 16.8 | 27.5 | 81.8 | 14.0 | 23.9 |
| LATEM Xian et al. (2016) | 57.3 | 15.2 | 24.0 | 28.8 | 14.7 | 19.5 | 71.7 | 7.3 | 13.3 | 77.3 | 11.5 | 20.0 |
| EZSL Romera-Paredes & Torr (2015) | 63.8 | 12.6 | 21.0 | 27.9 | 11.0 | 15.8 | 75.6 | 6.6 | 12.1 | 77.8 | 5.9 | 11.0 |
| SYNC Changpinyo et al. (2016) | 70.9 | 11.5 | 19.8 | 43.3 | 7.9 | 13.4 | 87.3 | 8.9 | 16.2 | 90.5 | 10.0 | 18.0 |
| DeViSE Frome et al. (2013) | 53.0 | 23.8 | 32.8 | 27.4 | 16.9 | 20.9 | 68.7 | 13.4 | 22.4 | 74.7 | 17.1 | 27.8 |
| f-CLSWGAN Xian et al. (2018b) | 57.7 | 43.7 | 49.7 | 36.6 | 42.6 | 39.4 | 61.4 | 57.9 | 59.6 | 68.9 | 52.1 | 59.4 |
| CVAE Mishra et al. (2017) | – | – | 34.5 | – | – | 26.7 | – | – | 47.2 | – | – | 51.2 |
| SE Verma et al. (2017) | 53.3 | 41.5 | 46.7 | 30.5 | 40.9 | 34.9 | 67.8 | 56.3 | 61.5 | 68.1 | 58.3 | 62.8 |
| ReViSE Tsai et al. (2017) | 28.3 | 37.6 | 32.3 | 20.1 | 24.3 | 22.0 | 37.1 | 46.1 | 41.1 | 39.7 | 46.4 | 42.8 |
| ours (CADA-VAE) | 53.5 | 51.6 | **52.4** | 35.7 | 47.2 | **40.6** | 72.8 | 57.3 | **64.1** | 75.0 | 55.8 | **63.9** |

Table 1: Comparing CADA-VAE with the state of the art. We report per class accuracy for seen (S) and unseen (S) classes and their harmonic mean (H). All reported numbers for our method are averaged over ten runs.

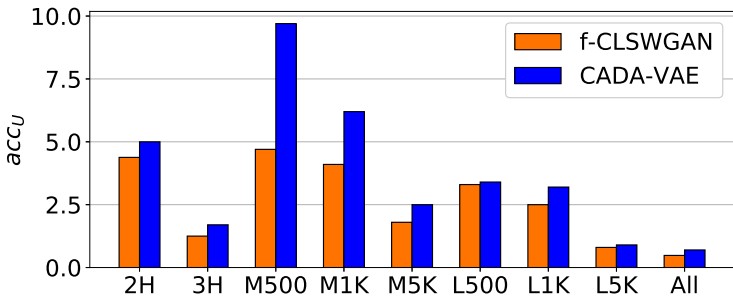

Figure 2: ImageNet results on GZSL. We report the top-1 accuracy for unseen classes. Both f-CLSWGAN and CADA-VAE use a linear softmax classifier.

act as distractors. For ImageNet, as attributes are not available, we use Word2Vec features as class embeddings provided by Changpinyo et al. (2016). We compare our model with f-CLSWGAN (Xian et al., 2018b), i.e. an image feature generating framework which currently achieves the state of the art on ImageNet. We use the same evaluation protocol on all the splits. Among the splits, 2H and 3H are the classes 2 or 3 hops away from the $1K$ seen training classes of ImageNet according to the ImageNet hierarchy. $M500$, $M1K$ and $M5K$ are the 500, 1000 and 5000 most populated classes, while L500, L1K and L5K are the 500, 1000 and 5000 least populated classes that come from the rest of the 21K classes. Finally, 'All' denotes the remaining $20K$ classes of ImageNet. As shown in Figure 2, our model significantly improves the state of the art in all the available splits. Note that the test time search space in the 'All' split is 22K dimensional. Hence even a small improvement in accuracy on this split is considered to be compelling. The achieved substantial increase in performance by CADA-VAE shows that our 128-dim latent feature space constitutes a robust generalizable representation, surpassing the current state-of-the-art image feature generating framework f-CLSWGAN.

## 4 CONCLUSION

In this work, we propose CADA-VAE, a cross-modal embedding framework for generalized zero-shot learning in which the modality-specific latent distributions are aligned by minimizing their Wasserstein distance and by using cross-reconstruction. This procedure leaves us with encoders that can encode features from different modalities into one cross-modal embedding space, in which a linear softmax classifier can be trained. We present different variants of cross-aligned and distribution aligned VAEs and establish new state-of-the-art results in generalized zero-shot learning for four medium-scale benchmark datasets as well as the large-scale ImageNet. We further show that a

cross-modal embedding model for generalized zero-shot learning achieves better performance than data-generating methods, establishing the new state of the art.

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
