# OpenReview forum: "Cross-Linked Variational Autoencoders for Generalized Zero-Shot Learning"
_ICLR.cc/2019/Workshop/LLD — LLD 2019_

### Official Review · AnonReviewer1 · 2019-04-07
**Interesting work on multi-modal generalized zero shot learning**

**Rating:** 4
**Confidence:** 3

**Review:**

Summary: The authors propose a VAE architecture that leverages multi-modal information, namely images and text in order to learn a matched latent space representation. The representation is created by minimizing two additional objectives in addition to the beta-VAE loss: the cross-reconstruction (CA) and the distribution-alignment (DA) regularizers. The learnt joint-posterior is then utilized to train a classifier in a GZSL setting. The method posed shows promising results as it beats SOTA in several benchmarks but should address the following points in the camera ready version.

Major:
  - There is no motivation or reasoning specified for the L1 cross-reconstruction (CA) loss in Equation 2. If the decoder is a gaussian then the natural assumption is a likelihood proportional to the L2 loss. If there is indeed a Laplace-likelihood this should be clearly stated. If not, then a small ablation study / discussion demonstrating the difference between the L1 & L2  (and corresponding distributional assumptions) for the CA should be provided.

  - For the classifier part of the model is the dimensionality of the softmax fixed? Or does it simply refer to which sample it is associated with as in few-shot learning? In addition, how is the classifier trained? I.e. does it use both \mu's and \Sigma's ? Are they concatenated? Passed through separate networks?

  - The final loss does not show the dependence on the hyper-parameters that weight the different terms of the loss; specifically the L_{CA} term seems to be hyper-parameter free? Training of multi-objective VAEs critically relies on the scale of these hyper-parameters. Beta in the beta-VAE is also not specified. These are critical and should be described.

Minor:
  - are the image features extracted from a pre-trained Resnet-101 model or is the encoder a Resnet-101 model? This should be made clear.
  - How many posterior samples are extracted for classification? How are they used? Is a classification made for each sample or is the latent representation averaged / concatenated and then classified? Why isn’t just the mean used as is standard in a test-setting for VAEs?
 - title is missing the word “Shot”

---

### Official Review · AnonReviewer2 · 2019-04-07
**Unclear writing with seemingly SOTA results on generalized zero-shot learning tasks**

**Rating:** 3
**Confidence:** 1

**Review:**

Pros:
- SOTA results

Cons:
- generally written in an unclear way
- Title should say "zero-shot"
- The indices of the covariance matrices in Fig 1 appear flipped
- Figure captions lack important information (for example, in Fig 1 there is no mention that the bottom VAE is for the class embeddings, and they also use c in the figure but c(y) in the text)
- not clear whether the results in Fig 2 for ImageNet are over multiple seeds, no error bars.

---

### Decision · Program_Chairs · 2019-04-08
**Acceptance Decision**

Accept